# Self-Organized Liver Microtissue on a Bio-Functional Surface: The Role of Human Adipose-Derived Stromal Cells in Hepatic Function

**DOI:** 10.3390/ijms21134605

**Published:** 2020-06-29

**Authors:** Seokheon Hong, Seung Ja Oh, Dongho Choi, Yongsung Hwang, Sang-Heon Kim

**Affiliations:** 1Center for Biomaterials, Biomedical Research Institute, Korea Institute of Science and Technology, Seoul 02792, Korea; heckham@naver.com (S.H.); seungja.oh@kist.re.kr (S.J.O.); 2Department of Surgery, Hanyang University College of Medicine, Seoul 04763, Korea; crane87@hanyang.ac.kr; 3Soonchunhyang Institute of Medi-bio Science (SIMS), Soonchunhyang University, Cheonan-si, Chungcheongnam-do 31151, Korea; 4Department of Bio-Med Engineering, KIST school, Korea University of Science and Technology, Seoul 02792, Korea

**Keywords:** multicellular hepatic microtissue, human adipose stem cell, induced hepatocyte, hypoxia-inducible factor 1α, hepatocyte maturation

## Abstract

The maintenance of hepatocyte function is a critical research topic in liver tissue engineering. Although an increasing number of strategies have been developed, liver tissue engineering using hepatocytes as a therapeutic alternative remains challenging owing to its poor efficacy. In this study, we developed a multicellular hepatic microtissue to enhance the function of induced hepatic precursor cells. Mouse induced hepatic precursor cells (miHeps) were self-organized in 3D with human adipose-derived stem cells (hASCs) on a bio-functional matrix. We found that hepatic phenotypes, such as levels of albumin, asialoglycoprotein receptor-1, and cytochrome P450, were enhanced in miHeps-hASC microtissue comprising miHeps and hASCs relative to two-dimensional-cultured miHeps-hASCs. Additionally, the secretome of 3D-cultured hASCs increased the hepatic function of mature miHeps. Furthermore, hepatic gene expression was reduced in mature miHeps treated with conditioned media of hypoxia-inducible factor 1α (HIF1α)-depleted hASCs relative to that with conditioned media of control hASCs. Our results suggested that the hepatic function of 3D-co-cultured miHeps could be enhanced by HIF1α-dependent factors secreted from stromal cells. This study provides an insight into the factors regulating hepatic function and shows that self-organized hepatic microtissue could act as liver spheroids for liver regenerative medicine and liver toxicity tests.

## 1. Introduction

Liver tissue engineering using hepatocytes is a therapeutic alternative for patients suffering from liver failure due to conditions such as hepatocirrhosis and as an in vitro model for studying drug development and preclinical phase evaluation [1,2,3]. Therefore, obtaining mature hepatocytes that exhibit high hepatic function is essential; however, techniques for culturing these cells remain an important challenge in liver tissue engineering.

Culturing large quantities of highly differentiated primary cultured hepatocytes that have high hepatic function remains challenging due to the loss of their ability to divide [4,5,6]. In order to solve this problem, various groups have attempted to differentiate pluripotent stem cells into mature hepatocytes with high proliferation and differentiation abilities [7,8,9]. It has recently been reported that somatic cells can be directly converted into parenchymal cells capable of proliferating, whilst direct conversion has also been shown to induce hepatic progenitor cells to obtain large numbers of mature hepatocytes [10,11]. Several studies have shown that regulatory genes upstream of hepatic phenotypes, such as HNF1α, HNF4α, Foxa1, Foxa2, and Foxa3, are involved in the transformation of fibroblasts isolated from patients into hepatic progenitor cells [12,13,14].

Tissue engineering enables biological tissue to be mimicked using cells, materials, and biochemical and physicochemical components [15,16,17]. To promote the function of engineered tissues, efforts have been made to culture cells in three dimensions (3D), generally by using a scaffold to support and grow cells into a 3D architecture [18]; however, recent studies have shown that cells can form 3D cellular tissues under scaffold-free culture conditions [19,20]. Previous studies have reported that hepatocytes maintain hepatic function when cultured in 3D spherical constructs [21,22], in particular when 3D co-cultured with other cells [23,24,25,26]. For instance, liver tissue-derived non-parenchymal cells can improve the survival and differentiation state of hepatocytes [27], whilst co-culture with stellate cells or sinusoidal endothelial cells can improve functional hepatic activity [23,25,26]. It has been reported that hepatocytes do not directly interact with the surrounding endothelial cells [26], but with specific cytokines secreted by Kupffer cells (e.g., tumor necrosis factor α (TNF α; pro-proliferative) and tumor growth factor β (TGFβ; anti-proliferative)) and hepatic stellate cells (e.g., hepatocyte growth factor (HGF)), which regulate their proliferation and maturation [23]. Furthermore, the peripheral extracellular matrix (ECM) environment created by these cytokines helps to maintain liver function and encourages the self-organization of 3D spheroids [23,25].

Human adipose-derived stem cells (hASCs) are multipotent stromal cells isolated from adipose tissue that can self-renew during culture and differentiate into various cell types, such as osteoblasts, chondrocytes, muscle cells, and adipocytes [28]. hASCs are widely used as a potent cell source to provide paracrine effects in tissue engineering and stem cell therapy [29,30]. Studies have shown that hASCs can form multicellular aggregates under various culture methods, such as the hanging-drop, spin vessel, and surface patterning methods [31,32,33]. Previously, we developed a unique material for culturing stromal cells, such as fibroblast and adipose-derived mesenchymal stem cells (ASCs), in 3D spherical structure [34,35]. hASCs could spontaneously aggregate into 3D spherical shapes on a polystyrene (PS) surface immobilized with basic fibroblast growth factor (FGF2) bound to maltose binding protein (MBP). MBP has a hydrophobic domain that can immobilize FGF2 on a hydrophobic PS surface. FGF2 specifically binds to heparan sulfate proteoglycans in the cell membrane of ASCs, causing the cells to adhere weakly to the surface and form a 3D structure [36]. We suggest that the biophysical balance of strength between cell contraction and cell-matrix adhesion might explain the mechanism of cell aggregation.

In this study, we produced a 3D multicellular hepatic microtissue by co-culturing hASCs and mouse induced hepatic precursor cells (miHeps), which were directly converted from mouse fibroblasts, on MBP-FGF2 surfaces. We examined the effect of hASCs and 3D culture conditions on the properties of miHeps in hASC-miHep co-spheroids.

## 2. Results

### 2.1. Hepatic Phenotype of Mouse Induced Hepatic Progenitors

To obtain the large amounts of mature hepatocytes required for liver tissue engineering studies, we used miHeps derived from primary cultured E16.5 mouse embryonic fibroblasts (MEFs) via direct conversion by introducing two hepatocyte-specific upstream regulators, Foxa3 and HNF4α, using the pMX viral vector [11,37]. To verify the hepatic phenotype of the miHeps, we analyzed their morphology and hepatocyte-specific marker gene regulation using RT-PCR, immunostaining and Western blotting during maturation. The miHeps divided repeatedly during cultivation with growth media, resulting in confluent cell monolayers on the culture dishes (Figure 1A, DIV2-6). Since miHep morphology changed during long-term culture with maturation medium containing DMSO (Figure 1A, DIV8-36), we analyzed the changes in hepatocyte-specific gene expression at a given time by RT-PCR. The expression of the hepatocyte-specific genes AFP, ALB, ASGR1, CYP450 (CYP1a2, CYP2a5, CYP3a13), and HNF4α increased significantly depending on incubation time in the monolayered culture with maturation medium (Figure 1B). Moreover, Western blotting and immunostaining analyses showed that the protein expression of ALB, ASGR1, CYP1a2, and E-cad, which are produced by mature hepatocytes, gradually increased with increasing culture time (Figure 1C,D). Furthermore, the membrane proteins ASGR1 and E-cad were localized on the surface of miHeps cultured in maturation medium for 36 days (DIV36) (Figure 1E), suggesting that miHeps are proliferative and can mature into hepatocyte-like cells during long-term culture. As shown in Figure 1F, miHeps cultured for eight days (DIV8) in growth and maturation media were positive for Ki-67, a cell proliferative marker, and rarely positive for albumin, thus being immature miHeps. When immature miHeps were cultured for a further 28 days in maturation medium, they were rarely positive for Ki-67, but positive for albumin, thus being mature miHeps.

### 2.2. hASC-miHep Co-Spheroid Formation and Morphological Characteristics

To acquire hASC-miHep co-spheroids reproducibly, we used MBP-FGF2-coated surfaces (PS-MBP-FGF2) that allowed hASCs to form a 3D spherical construct (Figure 2A). hASCs formed spheroids on PS-MBP-FGF2 in Cefo-gro^TM^ or DMEM/F12 (media for the maintenance of hASCs or miHeps, respectively) and their (1:1) mixture (Figure 2B); however, miHeps did not form spherical constructs by themselves on PS-MBP-FGF2 (Figure 2C). Reproducible co-spheroid formation was achieved by mixing 4 × 10^4^ hASCs and miHeps (1:1) on PS-MBP-FGF2 in a 96 well microplate, but not using the hanging-drop method (Figure 2D). As shown in Figure 2C, hASC-miHep co-spheroids formed within two days after culturing the hASCs and miHeps on PS-MBP-FGF2 using a 1:1 mixture of Cefo-gro^TM^ and DMEM/F12. The surface structure of the hASC-miHep co-spheroids was analyzed using SEM, showing that miHeps formed a cluster that appeared to be surrounded by hASCs (Figure 2E). Remarkably, microvilli-like structures were observed on the surface of miHeps aggregated in the co-spheroid (Figure 2E, middle and right images), but not in hASC spheroids (Figure 2E, left image). Microvilli are typically present on the surface of mature hepatocytes when their cell-to-cell interactions are enhanced [38], suggesting that the hASC–miHep represented 3D spherical forms of mature hepatocytes. To examine the localization of miHeps and hASCs inside the co-spheroids, they were labeled with CMTPX (red) and CMFDA (green), respectively, and observed using a confocal microscope. As shown in Figure 2F, clusters of green and red were observed in confocal cross-section images (Figure 2F, confocal plane) and the surface view of confocal stacked images (Figure 2F, stacked and computed). Taken together, these results indicated that mature and immature miHeps tended to self-aggregate in the inner and surface regions of co-spheroids.

### 2.3. Effects of hASC Co-Culture Conditions on the Hepatic Function of hASC-miHep Co-Spheroids

To examine the effect of 3D co-culture on the hepatic function of miHeps, we produced three types of co-spheroids with immature miHeps using growth or maturation medium and mature miHeps using maturation medium (Figure 3). qRT-PCR and RT-PCR analyses were performed to compare hepatic gene expression in hASC-miHep co-spheroids, 2D-cultured miHeps, and 2D-co-cultured hASC-miHeps. qRT-PCR analysis revealed that the expression of liver-specific genes, such as AFP, ALB, and ASGR1, decreased progressively when immature miHeps were co-cultured with hASCs under 2D and 3D conditions with growth medium compared to 2D-cultured miHeps (Figure 3A; immature/growth medium (GM)). Conversely, when immature and mature miHeps were co-cultured with hASCs under 2D and 3D conditions with maturation medium (Figure 3A; immature/ maturation medium (MM) and mature/MM) AFP, ALB, and ASGR1 expression increased compared to the 2D-cultured miHeps. Quantitative analysis of liver-specific mRNA expression, such as AFP, ALB, ASGR1, and CYP1a2, by RT-PCR showed that liver-specific genes were strongly expressed in mature miHeps 3D-co-cultured with hASCs in maturation medium (Figure 3A, right, A&H 3D and Figure 3B, mature/MM, dark gray). To verify the hepatic phenotype of each type of hASC-miHep co-spheroid, we performed immunostaining analysis for liver-specific proteins, such as ALB, ASGR1, CYP1a2, and E-cad (Figure 3C). Immunostaining analysis showed that all proteins were expressed more highly in mature hASC- miHep co-spheroids cultured in maturation medium than in the other co-spheroids. These results indicated that the expression of liver-specific genes in miHeps that induced their growth or maturation was greatly affected by hASCs and culture media. Thus, to increase the hepatic function of miHeps, the cells must be appropriately mature and 3D co-cultured with hASCs in maturation medium.

### 2.4. Effects of the hASC Secretome on the Hepatic Phenotype of hASC-miHep Co-Spheroids

hASCs secrete various factors that regulate the cellular activity of neighboring cells, such as cell proliferation and differentiation [39,40,41]. To investigate whether changes in liver-specific gene expression in hASC-miHep co-spheroids were due to factors secreted from hASCs, mature miHeps were cultured with the CM of hASCs cultured in 2D or 3D (Figure 4A). Immature miHeps grown in growth medium served as a control. No morphological changes were observed in either immature or mature miHeps cultured with the CM of 2D- or 3D-cultured hASCs (Figure 4B; 2D-CM or 3D-CM, respectively). The expression of liver-specific genes, such as AFP, ALB, and ASGR1, was analyzed using RT-PCR, showing that the hepatic gene expression of mature miHeps increased when treated with 3D-CM rather than 2D-CM, whereas that of immature miHeps decreased when treated with 3D-CM rather than 2D-CM (Figure 4C, left and middle). ALB and ASGR1 mRNA expression in primary hepatocytes increased when treated with 3D-CM, as in mature miHeps (Figure 4C, right). To verify the effect of 3D-CM on hepatic function in miHeps, immunostaining analysis was performed for albumin in immature and mature miHeps (Figure 4D). The number of albumin-positive cells increased when mature miHeps were treated with 3D-CM; however, there was no significant change when immature and mature miHeps were treated with 2D-CM. Conversely, the number of Ki-67-positive cells significantly increased and the number of albumin-positive cells decreased slightly in immature miHeps treated with 3D-CM (Figure 4D,E, immature miHeps); however, there was no difference in the number of Ki-67-positive cells when the mature miHeps were treated with hASC-3D-CM (Figure 4D,E, mature miHeps). These results suggested that the secretome of 3D-cultured hASCs increased the hepatic function of mature miHeps; therefore, the secretome of hASCs may up-regulate hepatic function in mature hASC-miHep co-spheroids.

### 2.5. Effects of the Secretome of Other Stromal Cells on miHep Regulation

To examine whether the hepatic function of miHeps was regulated by other stromal cells, such as human dermal fibroblasts (hDFs) or human bone marrow-derived mesenchymal stem cells (hBMSCs), immature and mature miHeps were treated with the CM of 2D- or 3D-cultured hDFs and hBMSCs. Few morphological changes were observed in the immature or mature miHeps when cultured with hDF-2D-CM, hDF-3D-CM, hBMSC-2D-CM, or hBMSC-3D-CM (Figure 5A). AFB, ALB, and ASGR1 gene expression was higher in the mature miHeps treated with hDF-3D-CM or hBMSC-3D-CM than with hDF-2D-CM or hBMSC-2D-CM, respectively (Figure 5B,C, right panels); however, their expression was lower in immature miHeps treated with hDF-3D-CM or hBMSC-3D-CM than with hDF-2D-CM or hBMSC-2D-CM, respectively (Figure 5B,C, left panels). These results supported our finding that miHep hepatic function responds to hASC-CM. The growth factor array of hASC, hDF, and hBMSC lysates showed that the expression of various growth factors promoting cell proliferation and differentiation, such as EGF, FGF, HGF, IGF, PDGF, TGF, and VEGF, increased by 1.5~4 fold in 3D-cultured hASCs, hDFs, and hBMSCs compared to in those cultured in 2D (Figure 5D). Taken together, the hepatic function of miHeps may be upregulated by soluble factors released by 3D-cultured hASCs, hDFs, and hBMSCs.

### 2.6. Effects of HIF1α on the Hepatic Phenotype via the hASC Secretome

Previously, we reported that hypoxia-inducing factor 1α (HIF1α) expression was induced by hypoxia in 3D-cultured hASCs [42]. To examine whether hepatic function, which was enhanced by 3D-hASC-CM in mature miHeps, was affected by HIF1α-induced factors, hASCs were transfected with HIF1α siRNA (Figure 6A). No morphological differences were observed in the hASCs following the transfection of control or HIF1α siRNA, and siRNA-transfected hASCs consistently formed spheroids on PS-MBP-FGF2 (Figure 6B). The HIF1α gene was downregulated in cells transfected with siHIF1α compared to those transfected with control siRNA (Figure 6C). Downregulation of the HIF1α gene and VEGF protein (typically regulated by HIF1α) was maintained in siHIF1α-transfected hASCs during an additional 24 h of culture for 3D spheroid formation (Figure 6D), verifying that HIF1α expression was decreased by HIF1α siRNA transfection in hASCs. Similar to the results shown in Figure 4C, the expression of liver-specific genes, such as AFP, ALB, and ASGR1, was downregulated for mature miHeps treated with the CMs collected from siHIF1α-transfected hASCs that were cultured in both 2D and 3D, compared to the control group: CMs of siRNA-transfected hASCs. On the other hand, the expression of hepatic genes was upregulated for immature miHeps treated with the CMs collected from siHIF1α-transfected hASCs that were cultured in both 2D and 3D (Figure 6E,F). These results indicated that soluble factors secreted from hASCs enhanced the hepatic function of miHeps in a HIF1α-dependent manner.

## 3. Discussion

Multicellular liver spheroids have been developed to test the liver toxicity of drugs and to study liver regenerative medicine [43,44]. Obtaining mature hepatocytes with high hepatic function is essential for developing liver spheroids. The differentiation of embryonic stem cells into hepatocytes is challenging from an ethical point of view, whereas using induced pluripotent stem cells to induce the regeneration of target cells after differentiation is time consuming and costly. To overcome the limitation of using pluripotent stem cells to generate mature hepatocytes, we obtained miHeps via the direct conversion of MEFs to obtain large quantities of mature hepatocyte-like cells to form functional liver organoids. These miHeps can be easily expandable in their growth medium, and their hepatic function can be enhanced by using maturation medium. In this study, we verified that miHeps could be sub-cultured to at least 20 passages whilst maintaining their immaturity in growth medium (data not shown). The expression of liver-specific genes, such as ASGR 1 and CYP450, increased remarkably in miHeps cultured under maturation conditions (Figure 1), suggesting that hepatic progenitor cells induced by direct conversion could be used to obtain a large number of functionally mature hepatocyte-like cells for liver tissue engineering.

Although various culture methods, such as hanging drop, PDMS micro-well, forced floating, and agitation-based bioreactor methods, have been reported for 3D spheroid formation [31,32], these methods did not reproducibly produce co-spheroids with a consistent size, shape, or cell ratio. In addition, a few studies have reported that primary cultured hepatocytes can be cultured in 3D; for example, hepatoma cell lines such as HepG2 have been cultured in 3D spheroids, and mouse primary hepatocytes form spheroids on carbohydrate-immobilized surfaces; however, the size and shape of the spheroids was not controllable [45,46].

Previously, we developed 3D microtissue models for cell therapy and tissue regeneration using PS-MBP-FGF2, which promotes the spontaneous formation of self-organized spheroids [42,47]. Furthermore, to produce functional 3D hepatocytes with high reproducibility, rapidity (<24 h), and ease of handling, in this study, we utilized the PS-MBP-FGF2 matrix as an artificial matrix for the spontaneous formation of hASC-miHep co-spheroids. In this study, we observed that miHeps could not form a 3D spherical construct alone using either the PS-MBP-FGF2 matrix or the hanging drop method. However, when miHeps were cocultured with hASCs in PS-MBP-FGF2, almost all cells formed co-spheroids with little cell loss, and co-spheroids were formed in all PS-MBP-FGF2 384 microplate wells (data not shown), suggesting that the PS-MBP-FGF2 method is advantageous for reproducibly and effectively obtaining co-spheroids for cell biology studies.

Epithelial cells themselves do not easily form stable constructs without neighboring stromal cells, which provide a matrix for cell adhesion and contract forces in vitro and in vivo; however, epithelial cells such as hepatocytes weakly form 3D constructs via cell-cell interactions mediated by membrane junction proteins, such as E-cadherin [48]. Previously, we reported that epithelial cancer cells and endothelial cells did not form 3D aggregates in PS-MBP-FGF2 [47]. Similarly, in this study, miHeps alone hardly formed spheroids in PS-MBP-FGF2, which could possibly explain their inability to adhere onto the MBP-FGF2-coated PS surface or the absence of aggregation-mediated spheroid-forming conditions, such as under agitation and centrifugation. Interestingly, spheroid formation in PS-MBP-FGF2 was slightly delayed in the hASC-miHep mixture compared to the hASCs alone (Figure 2B). Co-spheroid formation was further delayed as the proportion of miHeps increased (data not shown); suggesting that the ratio of miHeps to hASCs could be an important parameter for reproducibly and stably obtaining co-spheroids. In addition, co-spheroid formation was delayed when mature miHeps were used compared to immature miHeps. The maturity of the stromal cell-originated miHeps affected the formation of hASC-miHep co-spheroids; thus, miHep interference with spheroid formation could suggest that they are differentiating from stromal cells into hepatocytes (epithelial cells). miHeps clustered with neighboring miHeps and were surrounded by hASCs in the co-spheroids (Figure 2C and Figure 3D). Generally, epithelial and stromal cell tissues are spatially separated during organ development in the body [49,50]. Hepatocytes, which are highly differentiated epithelial cells, maintain cell-to-cell adhesion with neighboring hepatocytes to improve their function in liver tissue [51]. Interactions between each miHep in the co-spheroid are thought to be necessary for enhancing and maintaining liver function.

Growing interest in 3D culture techniques has led to the development of models that mimic the function of living tissues. The advantages of 3D culture compared to 2D culture include increased cell-to-cell contact, cell-to-ECM adhesion, and the release of soluble factors. Cell-to-cell contact and cell-to-ECM adhesion are generally recognized as the major biological processes that reinforce the function of cells in 3D culture [52,53]. Recent studies have shown that soluble factors released from stromal cells, such as BMSCs, umbilical cord-derived mesenchymal stem cells, and hASCs, can regulate liver function [54,55]. Our results suggested that increasing the hepatic function of liver-mimicking tissue to in vivo levels requires the 3D culture of not only hepatocytes, but also supporting cells capable of secreting ECM components and diffusible factors that regulate cell binding and hepatic function. Indeed, we showed that when miHeps were co-cultured with hASCs (supporting cells), particularly under 3D conditions, hepatic function increased significantly (Figure 3A,B, mature/MM). Consistently, we verified that the levels of various growth factors were elevated in the CM of other 3D-cultured stromal cells, such as hBMSCs, hDFs, and hASCs (Figure 5D). We used immature and mature miHeps to examine the effect of hASCs on the hepatic function, growth, and maturation of hASC-miHep co-spheroids. Immature hASC-miHep co-spheroids cultured in growth medium barely expressed hepatic phenotypes compared to 2D-co-cultured hASC-miHeps, whilst hepatic phenotype expression was markedly higher in mature hASC-miHep co-spheroids cultured in maturation medium. The hepatic phenotype of immature miHeps cultured with growth medium was reduced by adding 3D hASCs CM (Figure 4C); however, this also increased the growth activity of the immature miHeps (Figure 4D,E, Ki67 and ALB), explaining the inverse relationship between cell growth and maturation. The soluble factors secreted from 3D hASCs are an important aspect of the microenvironment that regulates miHep function and affects hepatic phenotype expression depending on the culture environment and miHep maturity. However, identifying the role of each growth factor that is produced more in 3D-cultured hASCs than those in 2D culture remains challenging.

Previous studies have reported that liver regeneration is induced by hypoxia: HIF1α stabilization protects hepatocytes from apoptosis via the Wnt signaling pathway, and hypoxia-induced VEGF production triggers sinusoidal endothelial cell proliferation during liver regeneration [56,57]; whilst HIF1α activity increases during liver regeneration, inducing a protective response and accelerating liver regeneration [58]. We demonstrated that hypoxic microenvironments were commonly introduced when culturing stromal cells in 3D [42]. Our current study showed that 3D-hASCs-induced hypoxic conditions and VEGF, a growth factor regulated by HIF1α, were downregulated by HIF1α knockdown (Figure 6D). In addition, protein array analysis revealed that insulin-like growth factor-binding protein-1 (IGFBP-1) levels were reduced in the CM of 2D- or 3D-cultured hASCs transfected with HIF1α siRNA (data not shown). IGFBP-1, which is expressed in the liver and binds IGF-1, is regulated by HIF1α in HepsG2, a human liver cancer cell line [59]. We examined whether 3D-hASC CM regulated the hepatic phenotype of miHeps via HIF1α, finding that HIF1α potently increased liver-specific gene expression in mature miHeps (Figure 6). Therefore, the hepatic function of miHeps seems to be regulated by the orchestration of HIF1α-dependent downstream factors, such as VEGF and IGFBP-1, alongside several other growth factors. Moreover, the inverse relationship between hepatic phenotype and hASC CM treatment, which depends on miIFS maturity, was shown to be equivalent in experiments using HIF1α-KD-hASCs (Figure 6). Although we did not demonstrate the role of each growth factor in the cellular response of miHeps, studies on these soluble factors could help determine the optimum culture conditions for developing in vitro liver models and establishing strategies to treat liver dysfunction.

## 4. Materials and Methods

### 4.1. Cells and Reagents

hASCs and human dermal fibroblasts (hDFs) were obtained from Cefobio (Seoul, Korea). miHeps and mouse primary cultured hepatocytes were obtained from Hanyang University (Seoul, Korea). Human bone marrow-derived mesenchymal cells (hBMSCs) were purchased from the Stem Cell Center of Catholic University (Seoul, Korea). All other reagents were purchased from Sigma-Aldrich, unless otherwise stated.

### 4.2. Cell Culture

hASCs (passage 5) were cultured under a 5% CO_2_ atmosphere at 37 °C in hASC growth medium (Cefobio, Korea) supplemented with 1% penicillin/streptomycin that was replaced every three days. hDFs and hBMSCs were maintained in high glucose DMEM and low glucose DMEM (Gibco, Gaithersburg, MD, USA), respectively, supplemented with 10% fetal bovine serum (Gibco, Gaithersburg, MD, USA) and 1% penicillin/streptomycin (Gibco, Gaithersburg, MD, USA), which was changed every other day.

miHeps were maintained at 37 °C in a 5% CO_2_ incubator on 250 μg/mL collagen type I (StemCell Technologies, Inc, Vancouver, BC, Canada) pre-coated plates in growth medium (DMEM/F12 with Glutamax (Gibco, Gaithersburg, MD, USA) supplemented with 10% fetal bovine serum (Gibco, Gaithersburg, MD, USA), 10 mM nicotinamide (Sigma Aldrich, St. Louis, MO, USA), 0.1 M dexamethasone (Sigma Aldrich, St. Louis, MO, USA), 1% insulin-transferrin-selenium-ethanolamine (ITS-X) supplement (Gibco, Gaithersburg, MD, USA), 1% penicillin/streptomycin (Gibco, Gaithersburg, MD, USA), 20 ng/mL HGF (PeproTech, Rocky Hill, NJ, USA), and 20 ng/mL epidermal growth factor (EGF; PeproTech, Rocky Hill, NJ, USA) that was changed every other day. For maturation, miHeps were cultured in growth medium for six days and growth medium containing 2% DMSO (maturation medium) for a further 28 days, with the media changed every other day. Mouse primary cultured hepatocytes were cultured under a 5% CO_2_ atmosphere at 37 °C in William’s E medium without glutamine (Gibco, Gaithersburg, MD, USA) containing cell maintenance supplements (Gibco, Gaithersburg, MD, USA) and 1% penicillin/streptomycin; half of the medium was replaced with fresh medium every three days during culture.

### 4.3. RNA Interference

Human HIF1α siRNA and negative control siRNA were purchased from Santa Cruz (SC-35561, California, CA, USA) and delivered (10 nmol) using the Lipofectamine RNAiMAX transfection reagent (Thermo Scientific, Waltham, MA, USA) according to the manufacturer’s protocol.

### 4.4. MBP-FGF2 Surface Preparation

MBP-FGF2 surfaces were prepared using our previously described method [38]. Briefly, MBP-FGF2 fusion proteins were obtained from *Escherichia coli* carrying pMAL-FGF2 plasmids that were generated by inserting human FGF2 cDNA (Bioneer, Korea) into pMAL vectors (New England Biolabs, U.K.). Human FGF2 cDNA was cloned from human fibroblasts via polymerase chain reaction (PCR) using oligonucleotide pairs (5′-CCG AAT TCC CCG CCT TGC CCG AGG ATG GC-3′ and 5′-CAA AGC TTT CAG CTC TTA GCA GAC ATT GGA AG-3′; Bioneer) with EcoRI and HindIII restriction sites, respectively. The PCR products were cloned into pGEM-T plasmids (Promega, Madison, WI, USA) to generate pGEM-FGF2. pGEM-FGF2 and pMAL plasmids were digested using EcoRI-HindIII, recovered from an agarose gel, and ligated using a ligation kit (TaKaRa, Shiga, Japan) to generate pMAL-FGF2. MBP-FGF2 (20 μg/mL) spontaneously adsorbed onto polystyrene (PS) surface plates (non-tissue culture-treated 96 or 384 well plates; Falcon, Fisher Scientific, Forest Lawn, NJ, USA) at 37 °C for 4 h.

### 4.5. Hepatic Microtissue Formation

hASC or hASC/miHep (1:1) suspensions (4 × 10^4^ cells) were seeded on an MBP-FGF2-coated 96 well PS plates in (1:1) supplemented mixed medium (hASC growth medium (Cefobio) and DMEM/F12 with Glutamax medium (Gibco, Gaithersburg, MD, USA)) and cultured for 24 h in an incubator at 37 °C under a 5% CO_2_ atmosphere. Microtissue was observed using an Axio Vert.A1 inverted microscope (Zeiss, Oberkochen, Germany).

### 4.6. Antibodies

The following antibodies were used: rabbit polyclonal anti-HNF4α (sc-8987, Santa Cruz, California, CA, USA), goat polyclonal anti-ALB (ab19194, Abcam, Cambridge, MA, USA), rabbit polyclonal anti-ASGR1 antibody (ab49355, Abcam Cambridge, MA, USA), mouse monoclonal anti-CYP1a2 (ab22717, Abcam Cambridge, MA, USA), rabbit polyclonal anti-β-actin (ab8227, Abcam Cambridge, MA, USA), mouse monoclonal anti-E-cad (ab76055, Abcam Cambridge, MA, USA), and rabbit polyclonal anti-Ki-67 (ab15580, Abcam Cambridge, MA, USA). The detailed information of primary antibodies used for immunoblotting assay and immunohistochemistry are summarized in Table 1.

### 4.7. Immunoblotting Assay

Immunoblotting was performed using 10^6^ cells in high salt lysis buffer [50 mM Tris-HCl pH 7.4, 10 % glycerol, 1 % Nonidet P-40, 300 mM NaCl, 150 mM KCl, 5 mM EDTA, 1 mM dithiothreitol, 5 mM NaF, 1 mM sodium orthovanadate, 10 μg/mL leupeptin, 10 μg/mL aprotinin, and 1 mM phenylmethylsulfonyl fluoride (PMSF)]. After incubation on ice for 20 min with agitation, the lysates were centrifuged at 10,000× *g* for 20 min at 4 °C, and equal volumes of dilution buffer were added to the supernatant (lysis buffer without NaCl and KCl) before boiling for 5 min at 94 °C in 2× SDS sample buffer (0.25 M Tris-HCl pH 6.8, 2% SDS, 10% 2-mercaptoethanol, 30% glycerol, and 0.01% bromophenol). Protein samples were separated by 10% SDS-polyacrylamide gel electrophoresis and transferred onto Immobilon P membranes (Millipore Corporation, Bedford, MA, USA). The membranes were blocked with 5% skim milk or bovine serum albumin in TBST (50 mM Tris-HCl pH 7.4, 150 mM NaCl, 0.1% Tween 20) for 1 h at room temperature and incubated overnight with primary antibodies (Table 1) in blocking solution at 4 °C on a rocker. After being washed three times with TBST for 10 min, the membranes were incubated with HRP-conjugated secondary antibodies (Jackson Immunoresearch, West Grove, PA, USA) for 1 h at room temperature. After washing three times, immunodetection was performed using a SuperSignal™ West Pico, enhanced chemiluminescence kit (Thermo Scientific, Waltham, MA, USA) according to the manufacturer’s protocol.

### 4.8. Semi-Quantitative Reverse Transcription Polymerase Chain Reaction 

Total RNA was extracted using an RNeasy Mini Kit (Qiagen, Valencia, CA, USA) according to the manufacturer’s instructions. First strand cDNA was synthesized by incubating 1 μg of total RNA at 70 °C for 5 min with 0.5 μg of oligo dT and deionized water (to make a total volume of 15 μL). Reverse transcription was performed using 200 units of M-MLV reverse transcriptase (Promega, Madison, WI, USA) in 5× reaction buffer (250 mmol/L Tris-HCl pH 8.3, 375 mM KCl, 15 mM MgCl2, and 50 mM DTT), 28 units of RNasin inhibitor, and 2.5 mM dNTP mixtures at 42 °C for 90 min with the following primers: HNF4α forward 5′-ATC GTC AAG CCT CCC TCT GC-3′ and reverse 5′-GAC TGG TCC CTC GTG TCA CAT C-3′; AFP forward 5′-AGC CTG AAC TGA CAG AGG AGC A-3′ and reverse 5′-TAA ACG CCC AAA GCA TCA CG-3′; ALB forward 5′-GGC TAC AGC GGA GCA ACT GA-3′ and reverse 5′-GCC TGA GAA GGT TGT GGT TGT G-3′; ASGR1 forward 5′-GGT GAC CTC CAG GGA TGA GCA GAA C-3′ and reverse 5′-CTG TTC CAT CCA CCC ATT TCC AGG GC-3′; CK18 forward 5′-GAC TGG GGC CAC TAC TTC AA-3′ and reverse 5′-CAT CTA CCA CCT TGC GGA GT-3′; CYP1a2 forward 5′- CAC AGC GAG AAC TAC AAA GAC AAT GGC-3′ and reverse 5′-GCT CCA GGT GAT GGC TGT GGT GAC-3′; CYP2a5 forward 5′-GCA CTT CCT AGA TGA CAA GGG ACA-3′ and reverse 5′-CAG GCT CAA CGG GAC AAG AA-3′; CYP3a13 forward 5′-TCC TGC AGA ACT TCA CTG TCC A-3′ and reverse 5′-TGG TTT CTG GTC CAC AGG ATA CA-3′; and GAPDH forward 5′-CCA ATG TGT CCG TCG TGG AT-3′ and reverse 5′- TTG CTG TTG AAG TCG CAG GAG-3′. A final volume of 25 μL with 2 μL of the cDNA products was used for each PCR reaction.

### 4.9. RT-PCR

RT-PCR was performed with master mix (TaKaRa, Shiga, Japan) at a final volume of 20 μL containing 10 μL of SYBR Green, 2 μL of each primer set used for qRT-PCR, 0.4 μL of ROX reference dye, and 20 ng of cDNA on a 7500 Real-Time PCR System (Applied Biosystems, USA) with the following thermal profile: 50 °C for 2 min and 40 cycles of 95 °C for 30 s and 15 s at 60 °C. Melting curve analysis was performed to detect primer dimerization affecting assay efficiency.

### 4.10. Immunocytochemistry

For immunostaining analyses, cells were dispersed and incubated on poly-L-lysine-coated 12 mm coverslips (Marienfeld, Lauda-Königshofen, Germany) in a 24 well plate, fixed with 4% paraformaldehyde in PBS for 5 min at room temperature, and permeabilized with PBS containing 0.5% Triton X-100 for 20 min. The cells were then blocked for 1 h using 2–5% normal serum in PBS containing 0.1% Triton X-100 and incubated with primary antibodies (Table 1) in blocking solution overnight at 4 °C. To visualize the primary antibodies, cells were incubated with Alexa Fluor (Molecular Probes, Life Technologies, Eugene, OR, USA) secondary antibodies for 40 min, washed, and mounted with mounting medium containing DAPI (ProLong^®^ Gold Antifade, Molecular Probes, Life Technologies, Eugene, OR, USA).

### 4.11. Cell Distribution Analysis

To produce hASC-miHep co-spheroids, hASCs and miHeps were stained with CellTracker dye (Thermo Scientific, Waltham, MA, USA) according to the manufacturer’s instructions. Briefly, hASCs and miHeps were incubated for 15 min with 1 μM 5-chloromethylfluorescein diacetate (CMFDA, green dye) or 1 μM chloromethyl 6-(4(5)-amino2-carboxyphenyl)-1,2,2,4,8,10,10,11-octamethyl-1,2,10,11-tetra hydrodipyrido[3,2-b: 20, 30-i] xanthylium (CMTPX, red dye), respectively. The cells were rinsed with PBS three times to completely remove the remaining dye, harvested, and incubated for microtissue formation for 48 h. Microtissues were fixed with 4% paraformaldehyde (PFA) for 5 min. Z-stack images were acquired using an LSM 700 laser scanning confocal microscope (Zeiss, Oberkochen, Germany), and the microtissue surface structure was reconstructed in 3D.

### 4.12. Scanning Electron Microscopy 

After washing with PBS, hASC-miHep co-spheroids were fixed in 2.5% glutaraldehyde solution at 4 °C for 30 min, maintained in 2% osmium tetroxide (OSO_4_) for 2 h at 4 °C, and washed with deionized water. After dehydration with dilute ethanol, hASC-miHep co-spheroids were dried by evaporation using hexamethyldisilazane (HMDS) and stored in a vacuum chamber for 24 h before imaging with an FE-SEM Hitachi S 4100 (Hitachi, Ibaraki, Japan).

### 4.13. Conditioned Media Assay

A total of 4 × 10^4^ cells were 2D- or 3D-cultured in 96 well tissue culture plates or MBP-FGF2-coated 96 well non-tissue culture plates, respectively. Conditioned media (CM) were obtained after 24 h, mixed with growth media or maturation media at a 1:1 ratio, and used to treat a confluent monolayer of miHeps in a 24 well tissue culture plate for 24 h.

### 4.14. Quantitative Densitometric Analysis

Scanned blot images were used for quantitative densitometric analysis using Adobe Photoshop CC (Adobe Systems Incorporated, San Jose, CA, USA). The mean intensity of a selected area around each band was calculated and used as a blank to determine the background of the bands. The pixels of the inverted selection area were then used to calculate band intensity. RGB and grayscale modes were used to obtain the absolute intensity of each band, with relative intensity calculated by normalizing against the control bands. Values were plotted on a relative scale in each graph.

### 4.15. Growth Factor Analysis

A human growth factor antibody array (ab134002; Abcam, Cambridge, U.K.) was used to investigate the growth factor expression profiles of each cell lysate. According to the manufacturer’s instructions, whole cell lysates (1 mL) were incubated overnight with membranes at 4 °C, washed with a large volume, and incubated with a biotin-conjugated anti-cytokine mixture overnight at 4 °C. Densitometry data were obtained using ImageJ (National Institutes of Health, Bethesda, MD, USA (http://imagej.nih.gov/ij/)) and used to compare different samples after background subtraction and normalization against positive control spots.

### 4.16. Statistical Analysis

All experiments were repeated at least three times and the results expressed as the mean ± SD of at least three independent experiments. Paired data were evaluated using Student’s *t*-tests. *p*-values of <0.05 (*), <0.005 (**), and <0.0005 (***) were considered significant.

## 5. Conclusions

In summary, we developed a novel hepatic spheroid using hASCs as supporting cells on a bio-functional surface. We proposed that hASCs not only provided a matrix for 3D miHep culture, but that their secretome enhanced the hepatic function of mature miHeps in co-spheroids. This study may help to produce hepatic spheroids with higher liver function; in particular, when using hepatocytes differentiated from stem cells to make 3D hepatic-stromal tissue constructs, hepatocyte maturity is crucial for creating an efficient liver model. Moreover, the secretome of 3D-cultured stromal cells could reinforce hepatic function and cell proliferation in the treatment of liver diseases. Hepatocyte-stromal cell co-spheroids have potential as an in vitro model for liver toxicity and disease studies and as cell therapeutics for liver regenerative medicine.

## Figures and Tables

**Figure 1 ijms-21-04605-f001:**
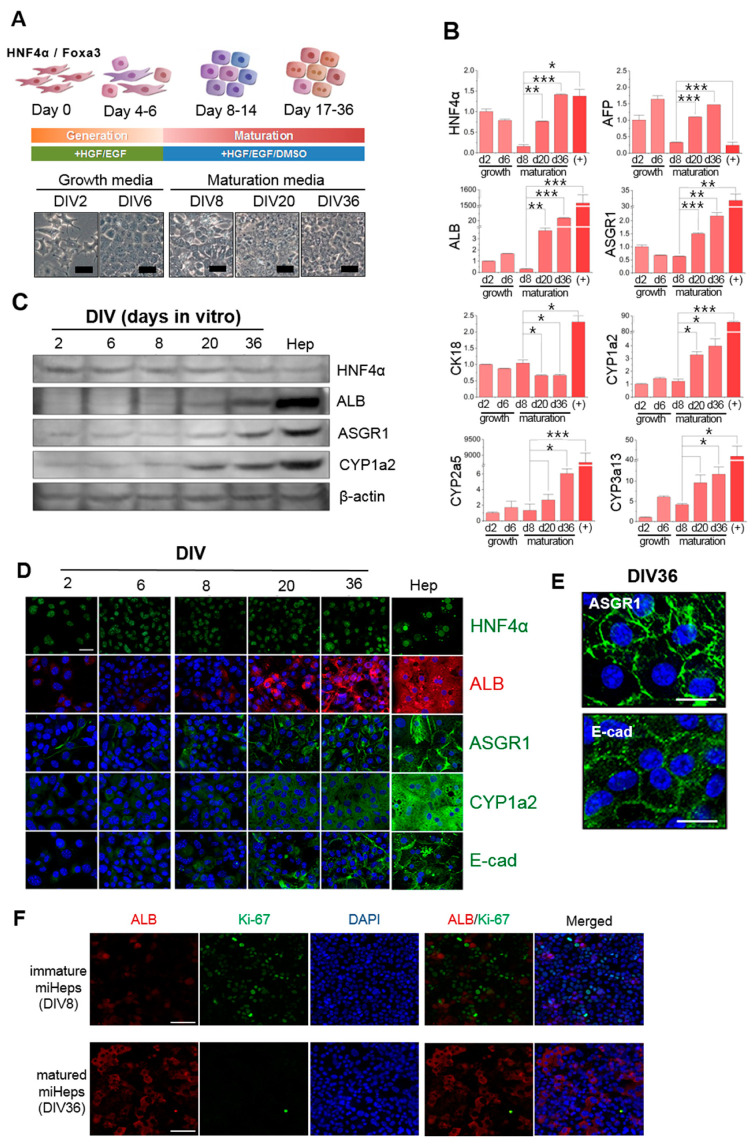
Maturation of mouse induced hepatic progenitors (miHeps). (**A**) Schematic overview of miHep maturation procedure (Scale bar: 50 μm). Real-time PCR (**B**) of the hepatic genes HNF4α, AFP, CK18, ALB, ASGR1, and CYP450s. Data represent the mean ± SD, * *p* < 0.05; ** *p* < 0.005; *** *p* < 0.0005. Immunoblotting (**C**) and immunocytochemistry (**D**) of the hepatic proteins HNF4α, ALB, ASGR1, CYP1a2, and E-cad in miHeps (Scale bar: 20 μm). Primary cultured mouse hepatocytes were used as a positive control. Magnified images of (**E**) ASGR1- and E-cad-stained 36 days in vitro (DIV36) miHeps (Scale bar: 20 μm) and (**F**) ALB- and Ki-67-stained immature and mature miHeps (Scale bar: 20 μm). All immunostaining (except HNF4α; a transduced factor) was carried out with DAPI (blue).

**Figure 2 ijms-21-04605-f002:**
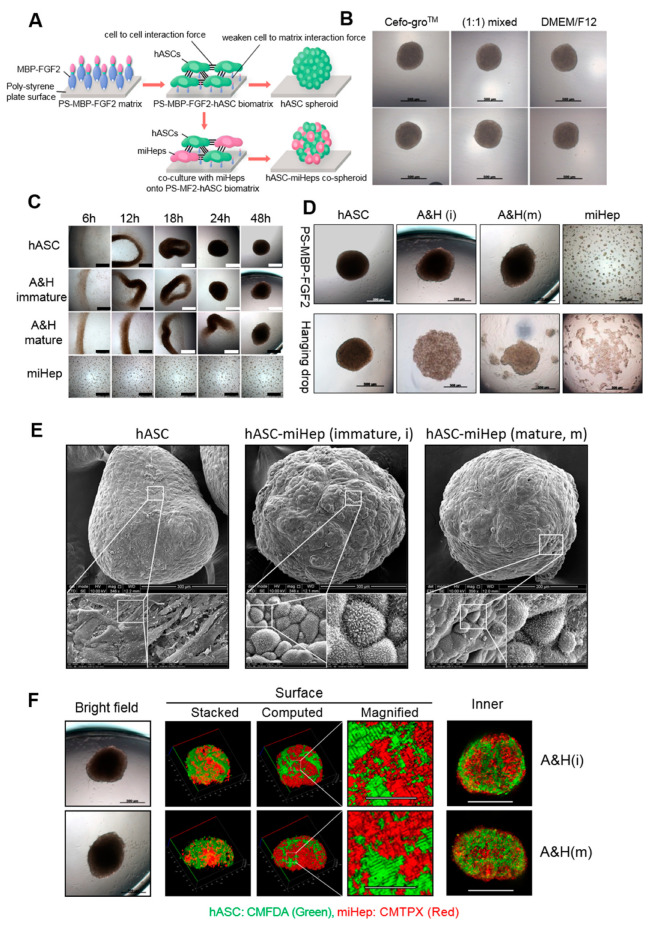
hASC-miHep co-spheroid formation and morphological characteristics. (**A**) Schematic overview of spheroid formation on PS-MBP-FGF2. (**B**) hASCs (1 × 10^4^ cells) were incubated with indicated media form hASCs spheroid in the MBP-FGF2-coated 384 well PS plate; the upper and lower images are duplicates observed by independent experiments. (**C**) hASC-miHeps co-spheroid formation. Immature or mature miHeps were co-cultured with hASCs to form co-spheroids using hASCs alone (hASC), miHeps alone (miHep), hASC-miHep (immature) [A&H (i)] and hASC-miHep (mature) [A&H (m)], respectively. Scale bar: 500 μm. (**D**) Cells (1 × 10^4^ cells) were cultured for 48 h in PS-MBP-FGF2 or by hanging drop using DMEM/F12:Cefo-gro^TM^ (1:1) media. (**E**) SEM images of spheroids: hASC spheroid (left), immature hASC-miHep co-spheroid (middle), and mature hASC-miHep co-spheroid (right). The magnified area is indicated with a white square. (**F**) Representative fluorescence images of immature (upper) and mature (lower) hASC-miHep co-spheroids (Left). Scale bar: 500 μm. Stacked images were analyzed to clarify the distribution of surface cells in co-spheroid and magnified (scale bar: 100 μm) images. Confocal plane images show the cell distribution inside the co-spheroids (scale bar: 500 μm).

**Figure 3 ijms-21-04605-f003:**
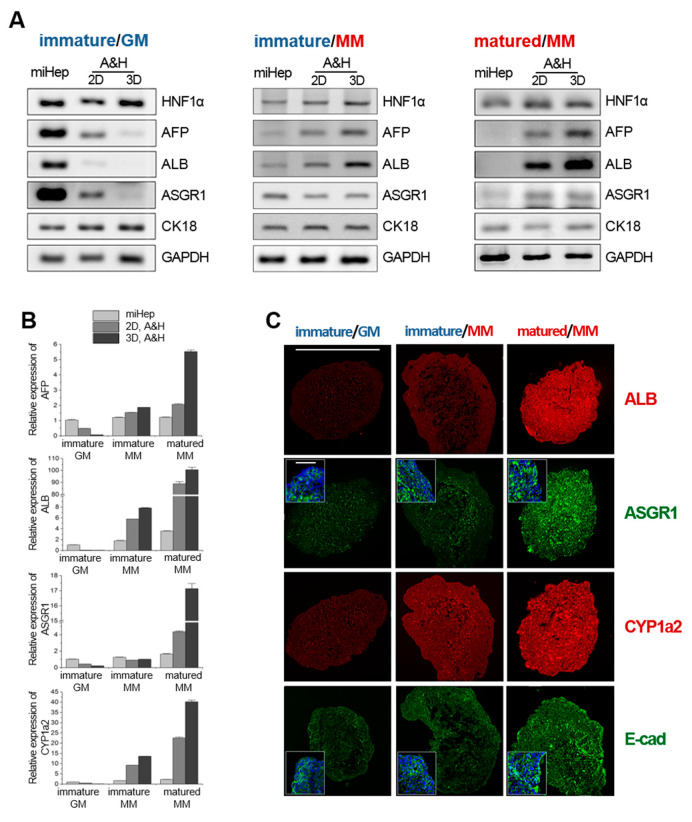
Effects of hASC co-culture on the hepatic function of hASC-miHep co-spheroids. (**A**) Reverse-transcription PCR and (**B**) quantitative real-time PCR analysis of miHeps and hASC-miHep co-spheroids; miHeps, 2D-cultured miHeps; immature, immature miHeps; mature, mature miHeps; GM, growth media; MM, maturation media. (**C**) Immunostaining analysis of co-spheroids (scale bar: 500 μm). The enlarged image distinguishes the membrane proteins (ASGR1 and E-cad; green) and nuclei (DAPI; blue) (scale bar: 50 μm). All measurements were taken 48 h after inducing spheroid formation.

**Figure 4 ijms-21-04605-f004:**
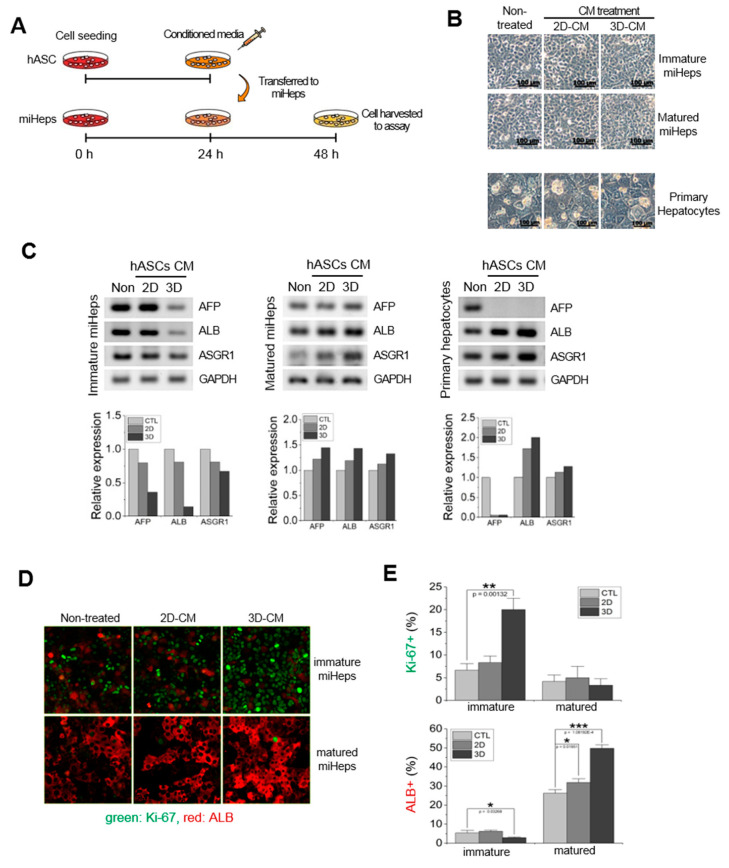
Effects of hASC conditioned media on miHep hepatic function. (**A**) Schematic diagram of the experimental procedure. (**B**) Cell morphology of miHeps and primary hepatocytes treated with CM; 2D-CM, the CM of 2D-cultured hASCs; 3D-CM, the CM of 3D-cultured hASCs. (**C**) Reverse-transcription PCR analysis of hepatic gene expression in hASC CM-treated miHeps and primary hepatocytes. All measurements were conducted 24 h after CM treatment. (**D**) Immunostaining of Ki-67 (green) and ALB (red) in immature and mature miHeps treated with hASC-CM. (**E**) Quantification of Ki-67+ and ALB+ cells. The percentage was normalized to the total cell count. Data represent the mean ± SD, * *p* < 0.05; ** *p* < 0.005; *** *p* < 0.0005.

**Figure 5 ijms-21-04605-f005:**
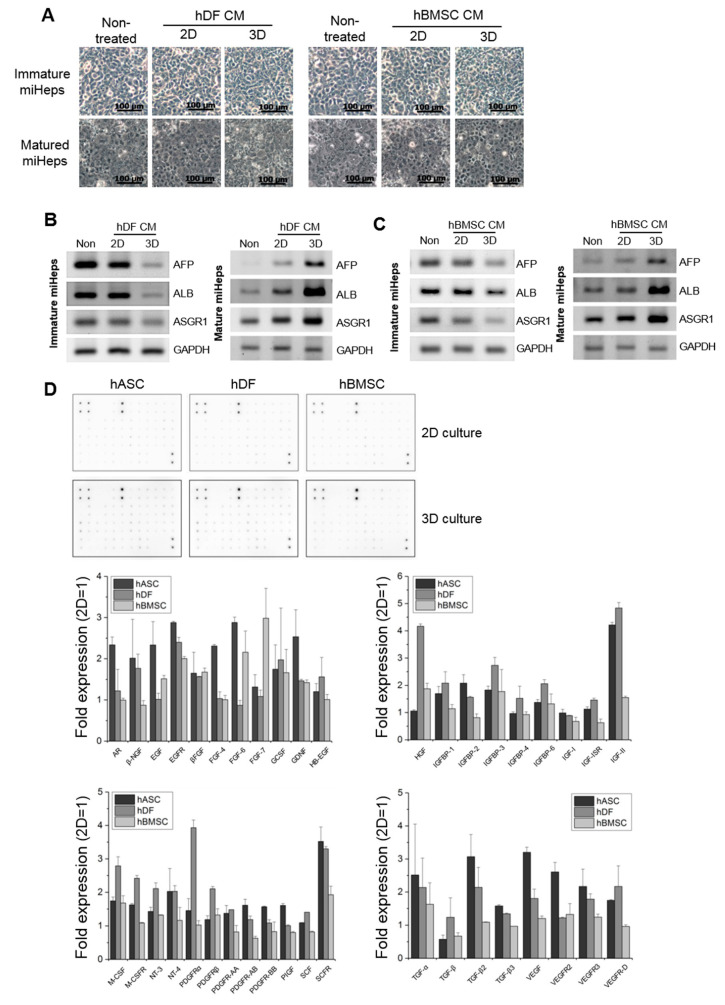
Effects of hBMSC and hDF conditioned media on miHep hepatic function. (**A**) Cell morphology of miHeps treated with the CM of 2D- or 3D-cultured hBMSCs and hDFs; 3D-CM, the CM of 3D-cultured hASCs. Reverse-transcription PCR analysis of hepatic genes in miHeps treated with the CM of hDFs (**B**) and hBMSCs (**C**). (**D**) Dot blot images were acquired from the lysates of 2D- or 3D-cultured hASC, hDF and hBMSC; The intensity of each dot blot was normalized with those of 2D-cultured cells set to one.

**Figure 6 ijms-21-04605-f006:**
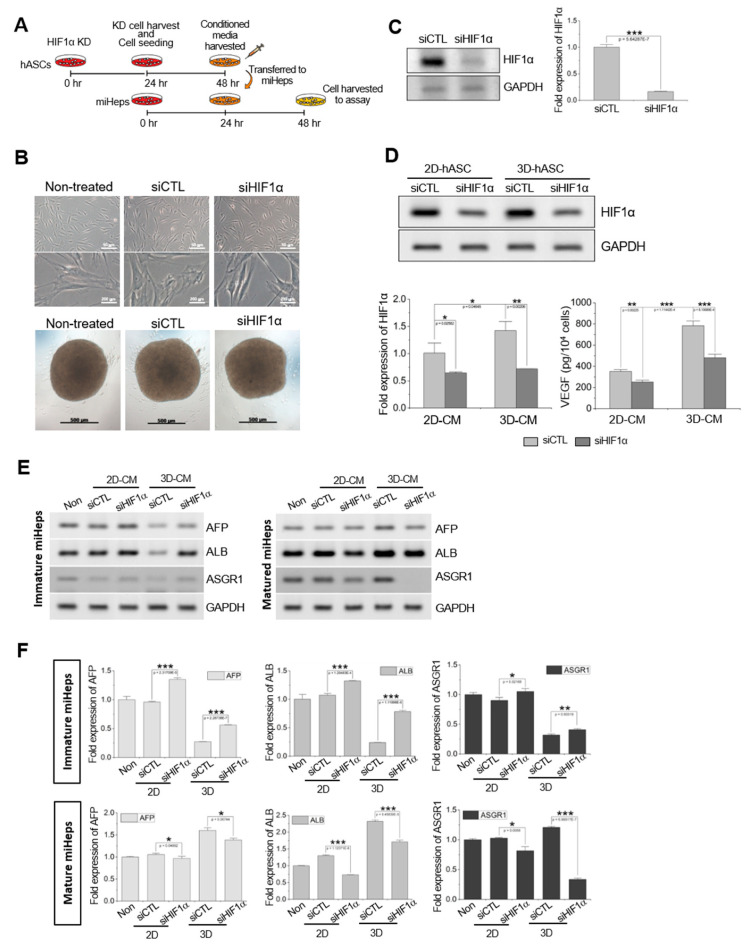
Effects of HIF1α on miHep hepatic function. (**A**) Schematic diagram of the experimental procedure. KD indicates knockdown. (**B**) Cell morphology of hASCs transfected with siRNA; siCTL, control siRNA; siHIF1α, HIF1α siRNA. Reverse-transcription PCR analysis of (**C**) HIF1α knock-down in hASCs 24 h after transfection and (**D**) 2D- and 3D-cultured hASCs 24 h after additional culture. Data represent the mean ± SD, * *p* < 0.05; ** *p* < 0.005; *** *p* < 0.0005. (**E**) Reverse-transcription PCR and (**F**) real-time PCR analysis of hepatic genes in miHeps treated with the CM of HIF1α-transfected 2D- or 3D-cultured hASCs. Data represent the mean ± SD, * *p* < 0.05; ** *p* < 0.005; *** *p* < 0.0005.

**Table 1 ijms-21-04605-t001:** List of primary antibodies used for immunoblotting assay and immunohistochemistry.

Antibodies	Dilution Factors	
	Immunoblotting	Immunohistochemistry
β-actin	1:2000	-
HNF4α	1:400	1:100
ALB	1:1000	1:500
ASGR1	1:500	1:200
CYP1a2	1:500	1:100
E-Cad	-	1:200
Ki-67	-	1:500

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
