# Peer review of "Self-Organized Liver Microtissue on a Bio-Functional Surface: The Role of Human Adipose-Derived Stromal Cells in Hepatic Function"

_ijms, 2020, doi:10.3390/ijms21134605_

Round 1
Reviewer 1 Report
The manuscript by Hong et al. describes a method to develop multicellular hepatic microtissue to enhance the function of induced hepatic precursor cells. Their study suggest that the hepatic function of 3D miHeps co-cultured with hASC could be enhanced by HIF1α-dependent factors. The findings from this study are of value for liver regenerative medicine.
However, there are a few additional examinations of the liver spheroids required to conclude usefulness of the model.
- The authors need to demonstrate the responsiveness to BMP9 the key regulator of liver progenitors.
- The study lacks functional assessments that are required for transplantation. Could the authors demonstrate the functionality of the liver spheroids by testing the indocyanine green (ICG) uptake/release or periodic acid‑Schiff staining.
- The authors need to demonstrate the responsiveness to commonly used liver damaging drugs by measuring hepatotoxicity-specific genes and ICG uptake.
- Could the authors state how long does their model hold good in exhibiting high levels of hepatocyte‑specific genes.
Author Response
Please find the attached rebuttal letter.

Reviewer 2 Report
Hong and colleagues established a 3D co-culture system of mouse induced hepatic precursor cells (miHeps) and human adipose-derived stem cells (hASCs) to analyse and improve hepatic properties of miHeps. The experiments are clearly described and straight forward.
1) Why did the authors used a co-culture system of a combination of murine and human cells? Why did they used human adipose-derived stem cells and not murine adipose-derived stem cells to stick to one organism?
2) The font size of Figure 2A is very small and difficult to read. I would suggest to increase the font size.
3) The results of the effects of Hif1a on hepatic phenotype via the hASC secretome are described in a very short way (page 12, line 243-247). I suggest to describe this in more detail and point also to the difference of immature and mature miHeps.
4) 4.3.RNA interference: Ask Santa Cruz for sequence details of the used siRNAs an include them.
5) 4.5.Hepatic microtissue formation: Was the experiment performed under 5% CO2 atmosphere?
6) 4.6. Antibodies: Provide the dilution of the single antibodies used for immunoblotting assays and immunohistochemistry.
Author Response

(The authors gave the same response as above.)
